# Balance between asymmetry and abundance in multi-domain DNA-binding proteins may regulate the kinetics of their binding to DNA

**Arumay Pal**, **Yaakov Levy** *

Department of Structural Biology, Weizmann Institute of Science Rehovot, Israel

* koby.levy@weizmann.ac.il

**Data Availability Statement:** All relevant data are within the manuscript and its Supporting Information files.

## Abstract

DNA sequences are often recognized by multi-domain proteins that may have higher affinity and specificity than single-domain proteins. However, the higher affinity to DNA might be coupled with slower recognition kinetics. In this study, we address this balance between stability and kinetics for multi-domain $Cys_2His_2$- ($C_2H_2$-) type zinc-finger (ZF) proteins. These proteins are the most prevalent DNA-binding domain in eukaryotes and $C_2H_2$ type zinc-finger proteins ($C_2H_2$-ZFPs) constitute nearly one-half of all known and predicted transcription factors in human. Extensive contact with DNA via tandem ZF domains confers high stability on the sequence-specific complexes. However, this can limit target search efficiency, especially for low abundance ZFPs. Earlier, we found that asymmetrical distribution of electrostatic charge among the three ZF domains of the low abundance transcription factor Egr-1 facilitates its DNA search process. Here, on a diverse set of 273 human $C_2H_2$-ZFP comprised of 3–15 tandem ZF domains, we find that, in many cases, electrostatic charge and binding specificity are asymmetrically distributed among the ZF domains so that neighbouring domains have different DNA-binding properties. For proteins containing 3–6 ZF domains, we show that the low abundance proteins possess a higher degree of non-specific asymmetry and vice versa. Our findings suggest that where the electrostatics of tandem ZF domains are similar (i.e., symmetrical), the ZFPs are more abundant to optimize their DNA search efficiency. This study reveals new insights into the fundamental determinants of recognition by $C_2H_2$-ZFPs of their DNA binding sites in the cellular landscape. The importance of electrostatic asymmetry with respect to binding site recognition by $C_2H_2$-ZFPs suggests the possibility that it may also be important in other ZFP systems and reveals a new design feature for zinc finger engineering.

## Author summary

Optimal recognition of proteins to DNA is governed by various factors among them the thermodynamics, kinetics and specificity of the protein-DNA complex. Multi-domain DNA-binding proteins are expected to have higher affinity and specificity due to the extensive interface they form with DNA. However, larger interface may result with higher

**Funding:** The author(s) received no specific funding for this work.

**Competing interests:** The authors have declared that no competing interests exist.

friction when these proteins scan the DNA for the target site via the sliding mechanism. A way to overcome this drawback is to have asymmetry in the protein so that the interface with DNA is smaller. Alternatively, higher abundance can also increase the search speed. Here, using computational analysis of large data set of multi-domain zinc finger DNA-binding proteins, we report a trade-off between asymmetry and abundance.

## Introduction

Multi-domain proteins are prevalent in eukaryotic systems and are involved in a variety of cellular functions[1,2]. The structural complexity of such proteins can assist in regulating binding via a network of protein–protein interactions. Multi-domain proteins that interact with DNA can be biologically useful to achieve higher specificity or tighter binding[3]. In many cases, cooperation between the tethered domains of multi-domain transcription factors were reported to be crucial for efficient binding to the DNA promoter[4,5]. The cooperation between the tethered domains can also support facilitated-dissociation mechanism from DNA [6,7].

Some multi-domain transcription factors were found to have asymmetrical non-specific binding affinities. For example, the two domains of the Oct1 transcription factor exhibit different non-specific binding affinities to DNA, with the N-domain binding much more tightly than the C-domain[8]. Similar asymmetrical binding affinity was found for the Pax6 transcription factor, however, in this case the C-domain interacts more tightly with DNA than the N-domain[8]. The non-specific binding affinity for these two proteins was estimated by assuming that it is determined solely by electrostatic interactions. Recently, asymmetrical non-specific DNA binding was found experimentally between the N- and C-terminal domains of the Pax5 transcription factor[9].

Similar asymmetry in non-specific binding affinity to DNA has been computationally predicted for other proteins[8], including between the Zinc Finger (ZF) domain constituents of several proteins. It was argued that this asymmetrical non-specific affinity is advantageous for the kinetics of DNA binding[10,11]. Given that genomic DNA includes numerous semi-specific binding sites, it takes time for a transcription factor to search the DNA for its target site. The presence of asymmetry in the non-specific binding affinities of two tethered domains allows the proteins to search DNA by transferring from one DNA segment to another via the monkey-bar mechanism, whereby each tethered domain interacts with different, distant DNA segments, thereby facilitating DNA search[12,13]. The monkey-bar mechanism is highly dependent on the asymmetry of the non-specific binding affinity.

The role of asymmetry in DNA search by multi-domain transcription factors was confirmed with respect to the monkey-bar search mechanism by an NMR study of the early growth response protein 1 (Egr-1) zinc finger, whose first finger has lower non-specific affinity to DNA than the two remaining fingers[10]. Mutating Egr-1 in a manner that increases the affinity of the first finger results in variants whose non-specific affinity is more symmetrically distributed[14]. This variant was found to transfer more slowly between distant DNA segments than the more asymmetric wild type. More importantly, it exhibited a lower rate of enzymatic cleavage of DNA when it was conjugated to a restriction endonuclease enzyme, FokI, compared with the faster enzymatic kinetics of wild-type Egr-1. This suggests that removing asymmetry in the distribution of non-specific DNA binding affinity across Egr-1's domains reduces the enzymatic cleavage rate of the FokI–Egr-1 conjugate[14].

Multi-domain transcription factors whose domains exhibit similar non-specific DNA-binding affinities (*i.e.*, that have symmetric binding affinities to non-specific DNA) are expected to have slower recognition kinetics for the specific DNA site. This slow kinetics may affect function. One may assume that not all transcription factors are required to recognize their target site rapidly and such cases may benefit from the multi-domain transcription factor exhibiting a more symmetric binding affinity and therefore overall higher specific affinity to DNA. Alternatively, multi-domain transcription factors that possess more symmetric non-specific binding affinity are expected to have faster recognition kinetics at their DNA target binding site when their cellular concentration is higher. Accordingly, multi-domain transcription factors having more symmetric non-specific binding affinity are expected to be more abundant in the cell to compensate for the lower probability of their performing monkey-bar dynamics when searching DNA.

We focus on the $Cys_2His_2$- ($C_2H_2$-) type ZF[15], which is the most prevalent human DNA-binding domain, constituting nearly 50% of all known and predicted transcription factors in human [16]. $C_2H_2$ ZF proteins (ZFPs) are involved in a range of nuclear processes including development, recombination, and chromatin regulation. $C_2H_2$-ZFPs are modular and typically have multiple ZF domains connected via short unstructured linkers to form arrays with lengths of up to 40 ZF domains.

In the canonical DNA recognition mode, each ZF domain usually interacts with a triplet of nucleic acid bases (3 bp) that are often recognized by four residues. Accordingly, the DNA binding specificity of each ZF domain is determined mostly by these four canonical 'specificity residues,' which are located at positions -1, +2, +3, and +6 on the recognition-helix[17,18] (Fig 1). Due to their unique mode of binding, $C_2H_2$-ZF domains are capable of specifically recognizing a wide range of 3 bp targets. Longer ZFPs often recognize the target gene using only a subset of the ZF domains in the array. Using multiple ZF domains in combination, $C_2H_2$-ZFPs can achieve remarkable diversity and specificity. They have also been subjected to various efforts to engineer specificity to DNA sequence[19]' [20]' [21,22]. Thus, a better understanding of the determinants of $C_2H_2$-ZF recognition of DNA would facilitate the design of engineered proteins with *de novo* binding specificities.

In humans, ZF binding specificities are known for less than a hundred of the approximately 700 $C_2H_2$-ZF proteins [23]. This reflects the complexity of determining them. Computational efforts have been made to characterize the interactions of $C_2H_2$-ZFPs with DNA sequences and to decipher their specificity. ZF domains are good targets for understanding biophysical concepts of protein–DNA affinity and specificity both because of the relatively simple interface they form with DNA and their large number. A comprehensive database containing information on individual $C_2H_2$-ZFs and engineered ZF arrays has been developed[24]' [25]. In addition to the DNA sequence and the identity of the residues at positions -1, +2, +3, and +6, the domain position in the array of fingers[26–28] and the interfaces between neighbouring fingers also affect binding specificity[26,29,30]. Using these principles, methods to predict DNA-binding specificities from sequence using amino acid-base pairwise energy have been reported [27,31,32]. Experimental studies highlighting the contribution of individual ZF domains to the collective specificity of the protein[33] as well as the role of DNA conformation in binding have also been reported [34].

This study focusses on multi-domain ZFPs because of the relative simplicity of their domains and the wide variety of ways in which they are used in the cell. Utilizing distribution of electrostatic charge between neighbouring ZF domains as a proxy for non-specific binding affinity and binding specificity score as an indicator of the specific DNA binding affinity, we examine the relationship between binding affinity asymmetry and cellular abundance for ZFP transcription factors.

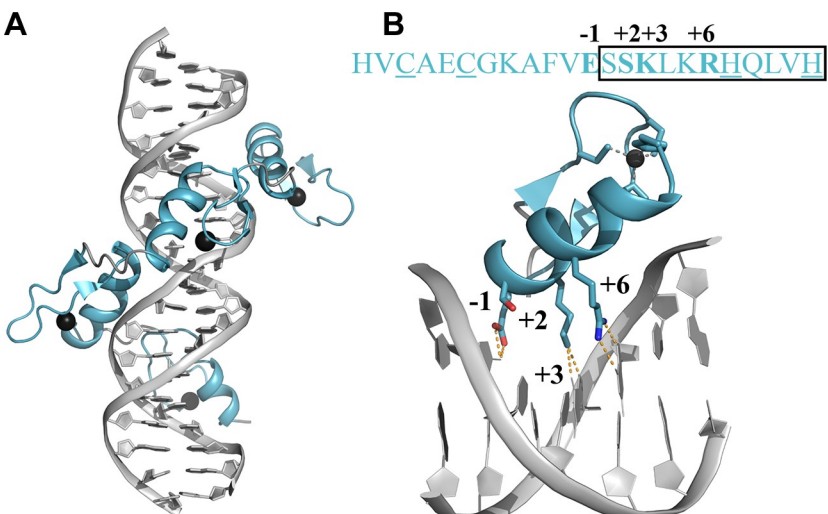

**Fig 1. DNA recognition by the C₂H₂-type zinc finger (ZF) domains of the human transcription repressor protein YY1.** (A) The crystal structure of the four ZF domains (blue) interacting with specific DNA (PDB code 1UBD; grey). Zinc atoms are represented by black spheres. (B) The amino acid sequence of the second zinc finger domain of the human transcription repressor protein YY1 (residues 325–347) is shown in blue letters at the top. The recognition helix is marked on the sequence by the rectangle. The two histidine and two cysteine residues that coordinate the zinc atom (black sphere) are underlined in the sequence. The four amino acid residues located at the four positions (-1, +2, +3, +6) involved in recognition by the protein of its specific DNA binding sites are indicated in bold in the sequence and represented by sticks in the 3D structure below it. The hydrogen bonds for the corresponding DNA base recognition are shown by dashed orange lines in the 3D structure.

## Materials and methods

### Dataset of zinc-finger proteins

The set of human $C_2H_2$ type ZF protein sequences used in this study was built by first searching the UniProt database[35] with the following queries: "annotation type–Zinc-finger", "organism–human", "existence–evidence at protein level," and "reviewed–yes". These queries yielded 477 $C_2H_2$-type ZFPs, in which the number of ZF domains in each protein varied immensely, from 1 (e.g., human protein arginine N-methyltransferase 3, Uniprot id: O60678) to 31 (e.g., zinc finger protein 142, UniProt id: P52746). Since a cluster of consecutive $C_2H_2$ ZF domains is thought to 'canonically' bind the major groove of DNA only when connected by short linkers[17,27], we further filtered to include only proteins whose ZF domains are connected by a linker shorter than nine residues. This filter produced a dataset of 237 unique ZFPs containing 3–15 $C_2H_2$-type ZF domains (S1 Table) and including 1911 ZF domains in total.

### Net charge on ZFs: Electrostatics as a proxy for non-specific binding affinity

In this study, we assume that the electrostatic interactions of ZF with DNA govern non-specific DNA binding by ZFs. Estimating the electrostatic contribution to the free energy of binding demands a high-resolution structure of the complex formed between the ZFP and the DNA. In the absence of bound structures for most of the ZFPs in the dataset with DNA, calculating electrostatic interaction energies from modelled complex structures is non-trivial, despite the great similarity of ZF structures. The complexity of predicting ZFP–DNA complexes for free energy calculations arises from the need to estimate the correct rotamers of the side chains at

the interface and also because the DNA may deviate from the canonical B-DNA geometry. Therefore, the electrostatic interactions between a ZF domain and a non-specific DNA sequence are estimated here by the net charge of the ZF. Given that the charged residues (Lys, Arg, His, Asp, Glu) are mostly present at surfaces and because we compare only ZF domains, we estimated the net charge from the sequence of each ZF domain by simply counting the difference between the number of positively (Lys, Arg, His) and negatively (Asp, Glu) charged residues. The net charge varied between -2 and +11; net charge $\leq 0$ was found for only $< 3\%$ of the total ZF domains in the dataset.

To compare the electrostatics of ZF domains from different proteins, we normalized the net charges of each ZF as $\bar{q}_i = q_i - \frac{1}{N-1}\sum_{j \neq i}^{N-1} q_j$ where, $\bar{q}_i$ is the net normalized charge and $q_i$ is the net charge of the $i^{\text{th}}$ ZF, $N$ is the total number of ZF domains in a ZFP with each component ZF domain numbered starting from the N-terminus.

## Specificity score as an indicator of specific DNA-binding affinity

To estimate the specificity of a given ZF domain, we developed a simple measure based on the propensity of each amino acid residue in our dataset of 1911 $C_2H_2$-type human ZF sequences to locate at the four 'specificity residue' positions (-1, 2, 3, and 6) in the DNA recognition helix. To that end, we first performed a multiple sequence alignment of these sequences by using the sequence of the first ZF (25 residues, positions 338–362) of human early growth response protein 1 (Egr-1) as the template. Other template sequences of different length (e.g., the third finger of transcription factor Sp1, 23 residues) provided similar propensity values. The maximal propensities at these four positions was found to be 18–20%, except for Ser at position +2 which exhibits a maximal propensity of 61%. Negatively charged residues, though present in the template sequence, are not typically found at all these positions, whereas a few occurrences of positively charged residues at positions +2 and +3 suggests that the interaction specificity of ZF proteins is not necessarily coupled with their binding affinity.

The multiple sequence alignment is used to estimate the probability, $P_x^y$, to find each of the 20 amino acids at each of the four positions (namely, for X = -1, +2, +3, or +6). Using these propensity values, the specificity score, $S$, for a given ZF can be estimated by

$$S = \ln(\prod P_x^y)/\ln(\prod P_x^{y,max})$$

where $P_x^{y,max}$ is the maximal propensity of residue $y$ to locate at the $x$ position (X = -1, +2, +3, or +6) in the query ZF sequence as calculated from the multiple sequence alignment. Propensity values are normalized by the maximal propensity to keep the specificity score in the range 0–1.

We note that the specificity with which a ZF recognises its DNA partner is dependent on the DNA sequence as well, so the ZF sequence alone cannot fully represent the specificity. However, our purpose in calculating the specificity of each ZF domain is to compare their values to examine whether some fingers are highly specific compared with the others, which indicates the presence of asymmetric specificity among different ZF domains in a ZFP. Hence, the specificity scores calculated from the ZF sequence alone remain informative here.

## Degree of asymmetry of non-specific and specific DNA-binding affinity

We estimate the degree to which non-specific binding affinities are asymmetrical in a given ZFP by calculating the percentage of electrostatically asymmetrical ZF pairs (where a ZF pair comprises two adjacent ZF domains, each bearing electrostatic charge). For example, a ZFP[3] has two pairs of consecutive ZF domains (ZF$^{1st}$–ZF$^{2nd}$ and ZF$^{2nd}$–ZF$^{3rd}$), so its percentage of

asymmetric pairs may be 0% (when neither pair of domains possesses electrostatic asymmetry), 50% (when either pair is asymmetric), or 100% (when both pairs are asymmetric). Similarly, a $ZFP^4$ has three pairs of consecutive ZF domains ($ZF^{1st}$–$ZF^{2nd}$, $ZF^{2nd}$–$ZF^{3rd}$, and $ZF^{3rd}$–$ZF^{4th}$) and thus can have 0%, 33.3%, 66.6%, or 100% asymmetry. A pair of neighbouring ZF domain is categorized as asymmetric if the difference between the net charges of the two neighbouring ZF domains is equal or larger a cutoff, $\sigma^{non\text{-}spec}$. A cutoff of 3e was used but other values gave similar results.

A similar approach to that used to estimate non-specific binding affinity was adopted to defining specific binding asymmetry, with the principal difference being the physical property considered, namely, binding specificity instead of electrostatics. Thus asymmetry in specific DNA binding affinity is considered to occur when either or both the following two conditions are satisfied: i) the specificity score of one of the ZFs is low ($< 0.6$), and ii) the difference in specificity score between two adjacent zinc finger domains (*i.e.*, within a zinc finger pair), cutoff of $\sigma^{spec} \geq 0.2$. Other cutoff values yielded similar results.

## Abundance

We used the available data on ZFP abundance provided by the absolute Protein Abundance Database (PaxDb: http://pax-db.org), which contains whole genome protein abundance information across organisms and tissues from experimental data [36]. These values are linearly proportional to protein copy numbers in cells.

## Results

### Electrostatic properties of human $C_2H_2$ zinc-finger domains

Positively charged residues (Lys, Arg, and His) are found to be distributed throughout the ZF sequence (S1 Fig). When mapped on the surface in the ZF structure, the charged residues are mostly found in the recognition helix; particularly, the probability of Lys, Arg, or His occurring at positions 13–15, 19–20, and 22 is between 20–30% and at position 23 is about 80%. However, positively charged residues are also found at the opposite side of the DNA binding surface, such as the β-strand (positions 2 and 10). Interestingly, negatively charged residues (Asp and Glu) are found only at the opposite side of the DNA-binding surface, mostly in the loop between the two Zn-coordinating Cys residues (positions 6–7) (Fig 1, S1 Fig).

From all the available $C_2H_2$-type ZF–DNA complex structures, it is known that ZF domains recognize DNA with a conserved canonical conformation in which the residues in the recognition helix interact with the DNA bases in the major groove. However, the distribution of charged residues on the ZF surface described above indicates that the number of positive and negative residues in individual ZF domains would dictate the overall strength of the long-range electrostatic attraction. A ZF domain with higher numbers of positive charges (high net charge) would have tighter non-specific DNA association compared with one with fewer positive charges (lower net charge).

The average values of the net charges appear to be similar for ZF domains from ZFPs of different lengths, with a slight greater net charge for shorter ZFPs of 3–6 ZF domains (Fig 2A). This may suggest that shorter ZFPs require greater electrostatic attraction to interact with DNA. Nevertheless, the difference in the net charge between shorter compared with longer ZFPs is of about a single charge. On average, an individual ZF domain contains 6.5 (± 1.4) positively charged and 1.6 (± 1.0) negatively charged residues, which gives it an overall average net positive charge of 4.9 (3.9 positive charges and 2.3 net charges, respectively, when His is not considered a positive residue). We note that some ZF domains have net charges of -2 or even +12, but these cases are rare.

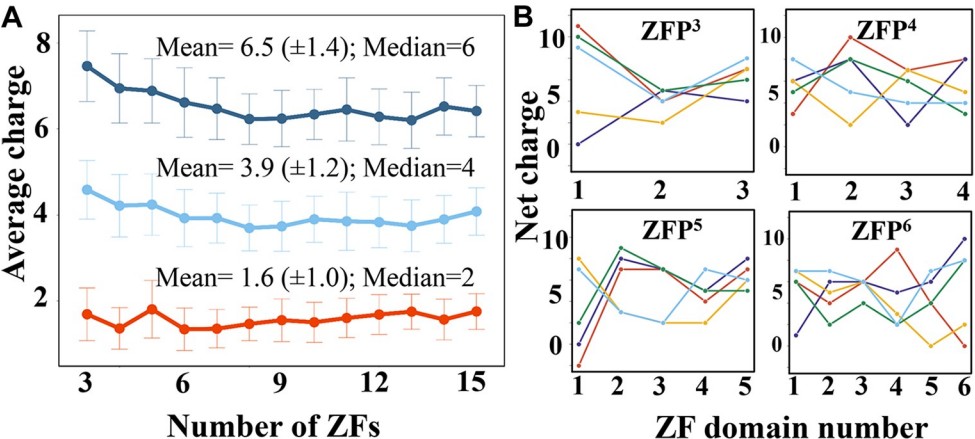

**Fig 2. Electrostatic properties of the C₂H₂-type zinc-finger proteins (ZFPs).** (A) The average number of positive (dark and light blue) and negative (red) charges are shown for human $C_2H_2$-type tandem ZFPs of different lengths (comprised of 3 ($ZFP^3$) to 15 ($ZFP^{15}$) zinc finger (ZF) domains. The His residue may be present in its positively charged (deep blue line) or neutral (light blue line) form. (B) The number of net charges on each zinc finger is shown for $ZFP^3$–$ZFP^6$. Each panel presents the analysis of five representative ZFPs (shown in five different colours) of the same length (as indicated by the superscript in the panel title). The analysis shows that the net charge of individual zinc finger domains within a protein can vary significantly.

We sought to examine the degree of variance in the net charge of individual ZF domains within a single ZFP. Fig 2B shows a few examples of the net charge in consecutive ZF domains in proteins comprising 3–6 ZFs, with the number of ZF domains indicated by superscript. (*i.e.*, $ZFP^3$, $ZFP^4$, $ZFP^5$, and $ZFP^6$). The figure illustrates the considerable variability in net charges in ZFPs. Variability is observed within each ZFP, in that the various ZFs may have high ($\geq 7$) and low ($\leq 2$) net charges (see within-panel variation in the net charge of each different coloured protein). Variability is also observed between different ZFPs (see between-panel variation in the net charge patterns for ZFP variants of different lengths).

## Electrostatic property variation in tandem zinc fingers in human C₂H₂ proteins

Variation in the net charge of tandem ZF domains can be quantified by a measure that estimates the degree of asymmetry in the electrostatics of neighbouring ZF domains. This measure indicates the percent of pairs of neighbouring domains that have different electrostatic characteristics. A ZF pair is counted as having asymmetric electrostatics if one of the ZF domains in the pair has a lower net charge (indicating a lower non-specific DNA affinity) whereas the other has a higher net charge (indicating a higher non-specific DNA affinity). Following this criterion, within each ZFP, we identified the ZF domain pairs whose constituent domains had asymmetrically distributed net charges and, thus, possessed electrostatic asymmetry. Since the purpose was to compare the net charges of the ZF domains within individual ZFPs, it was necessary to quantify the net charge of each ZF domain with respect to the others in the protein. Thus, we first normalized the net charge of each ZF domain by subtracting the average net charge of the other ZF domains in that ZFP from the net charge of the ZF domain under consideration (see Method). A pair of consecutive ZF domains was then considered electrostatically asymmetric if it satisfied both of the following conditions: i) the net charge of one of the ZF domains is negative ($< -2e$); and ii) the difference between the net charges of the two neighbouring ZF domains, $\sigma^{\text{non-spec}} \geq 3e$.

Fig 3 examines ZFPs comprising 3–6 ZF domains and shows the net charge on each domain. In these plots, each line corresponds to a single ZFP. The zigzag shape of some of the

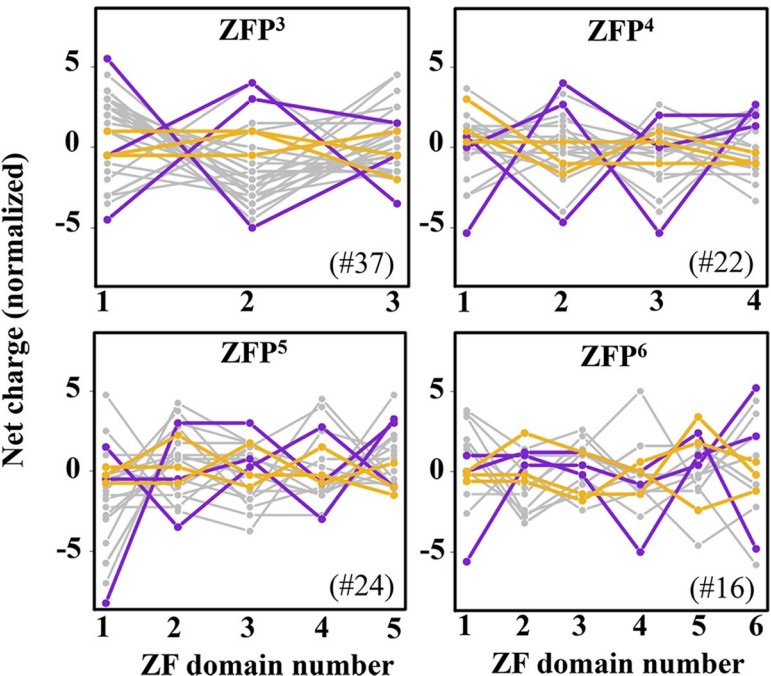

**Fig 3. Net charge of individual zinc-finger domains in tandem zinc-finger proteins.** The panels show the variation in normalized net charges for zinc finger proteins (ZFPs) comprising 3 (ZFP³) to 6 (ZFP⁶) zinc finger (ZF) domains. The number of domains is indicated by the superscript on ZFP in the title and the number of proteins shown in each group is indicated in the panel by a # mark. In each panel, three examples of proteins having asymmetric (purple) and symmetric (orange) electrostatics are shown. The normalized net charge of a ZF domain was obtained by subtracting the mean net charge of all the other ZF domains in that protein from the net charge of the ZF of interest.

lines indicates variation in the net charge between the tethered ZF domains in each ZFP and may correspond to asymmetrical electrostatics, whereas the more horizontal lines signify greater electrostatic symmetry between neighbouring ZF domains. To illustrate these two scenarios, Fig 3 displays three proteins that show considerable variation in the electrostatics of their constituent ZF domains in purple and three different proteins that exhibit milder variation in orange. The former cases have one or more electrostatically asymmetric ZF pairs and the latter have no asymmetric ZF pairs.

Fig 4A shows that shorter proteins (ZFP³ to ZFP⁶) contain a higher percentage of asymmetric than longer proteins. Yet, some ZFPs with a larger number of ZF domains were also classified as asymmetric. For example, 50% (6 of 12 pairs) of the pairs in one of the ZFP¹³ variants are asymmetric. Similar trends were obtained using a higher value for $\sigma^{non-spec}$ (4e, 5e, and 6e) as the second condition to define an asymmetric pair, as described above (S2 Fig). Overall, the data clearly indicate that non-specific asymmetry (namely, asymmetry in the net charge) is rather common in $C_2H_2$-type tandem ZF transcription factors, which may affect their DNA binding mechanism. On the other hand, many proteins, across all lengths, possess 0% asymmetry in charge (Fig 4A) leading to a different group where the lack of asymmetry in charge may lead them to adopt a different DNA binding mechanism.

## Variation in the specificity of tethered zinc finger domains

Apart from affinity, binding to DNA often requires specificity. Structural studies have revealed that each ZF domain in a $C_2H_2$-type ZFP binds a 3 bp region of DNA and that four amino acids in the recognition helix at positions -1, +2, +3, and +6 largely determining binding

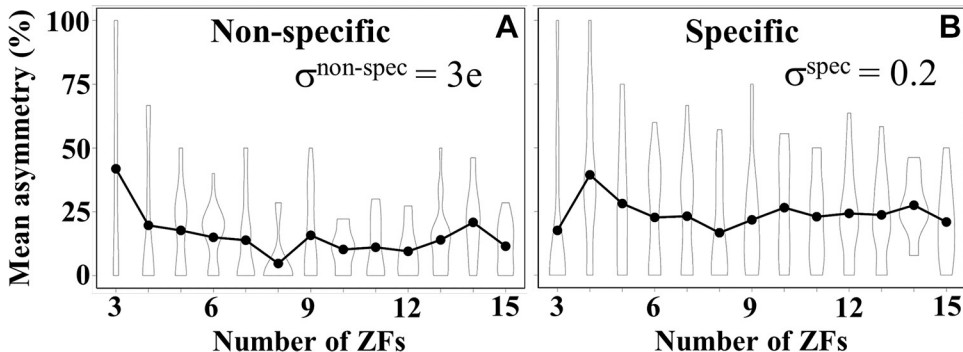

**Fig 4. Percentage of non-specific and specific binding asymmetry in zinc finger proteins (ZFPs) of different lengths.** Each dot represents a ZFP: the average percentage of asymmetric zinc finger (ZF) pairs for each length (*i.e.*, number of ZF domains) is shown by a solid circle, with the corresponding standard deviation shown by the error bar. (A) Asymmetrical non-specific binding was considered to occur when the difference in net charge between two neighbouring ZF domains was $\sigma^{nonspec} \geq 3e$. (B) As a course approximation, asymmetrical specific binding was considered to occur when the difference in specificity score between two neighbouring ZF domains was $\sigma^{spec} \geq 0.2$.

specificity. Amino acid side chains at these positions are generally involved in bonding as hydrogen bond donors or acceptors to form base-specific hydrogen bonds to recognize the DNA sequence. Therefore, residue propensities at these four specific positions as found in natural ZFP sequences can be used to predict the interaction specificity for a given ZF domain. Aligning all the ZF sequences in our dataset, the probability of finding each amino acid at these four positions was calculated. These four positions accommodate different residues, however they have a greater propensity to accommodate positively charged (Lys, Arg, His) and polar (Ser, Thr, Asn, Gln, His) side chains (S1 Fig). Site +2 has an exceptionally high preference for Ser (61%). Negatively charged residues do not typically occur at these positions (<8%), whereas the occurrence of Lys and Arg at the -1 and +6 positions is 22% and 31%, respectively. This may suggest that ZF specificity might be coupled to some extent with ZF binding affinity.

Here, the specificity score of a given ZF is estimated from the probability of finding it at those four specific positions, which is in turn dependent on its propensity to locate at those residue positions (see Method). The specificity value lies in the range 0.0–1.0 where a higher value indicates greater specificity. Fig 5 shows the average specificity scores for ZF domains from ZFPs of various length. It appears that ZFs from shorter ZFPs (comprised of 3–7 ZF domains) have lower specificity scores than those in longer ZFPs with 8–15 ZFs. However, all the average specificity scores are >0.5, suggesting that, in general, ZF domains use conserved side-chains at those four specific positions in the recognition helix. This is more pronounced in ZFPs possessing a larger number of ZF domains, whereas ZFPs with fewer ZF domains (ZFP³ to ZFP⁶) show variable specificity. Fig 5B highlights the specificity of individual ZFs for the same set of proteins shown in Fig 2B, demonstrating that variation exists in the specificity score of tethered ZF domains of the same protein.

The approach we adopted to defining specific binding asymmetry was consistent with the approach we used earlier to define non-specific binding asymmetry, with the principal difference being the physical property considered, namely, binding specificity instead of electrostatics. Thus asymmetry in specific DNA binding affinity is considered to occur when either or both the following two conditions are satisfied: i) the specificity score of one of the ZFs is low (< 0.6), and ii) the difference in specificity score between two adjacent zinc finger domains (*i.e.*, within a zinc finger pair), $\sigma^{spec} \geq 0.2$. Fig 4B shows the percentage of asymmetric pairs in ZFPs of length 3 to 15 domains (ZFP³ to ZFP¹⁵) when asymmetry is considered in terms of

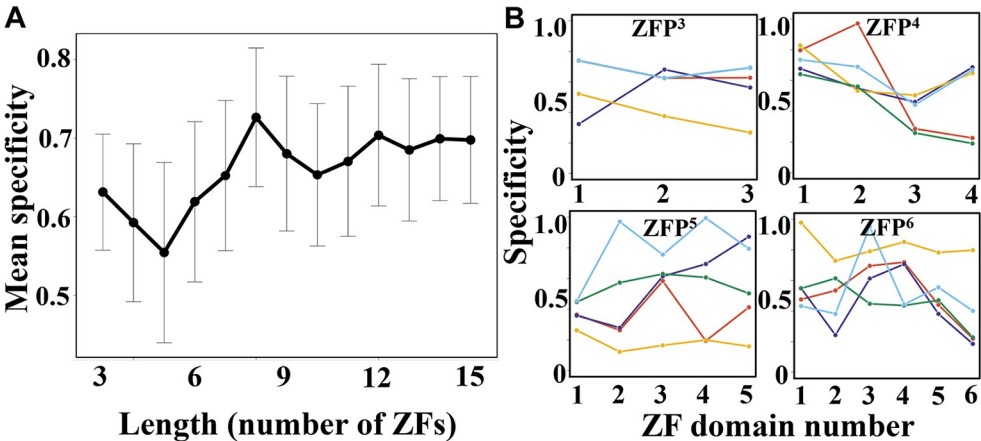

**Fig 5. Binding specificity in C$_2$H$_2$-type tandem zinc-fingers.** (A) Mean DNA binding specificity is shown for human C$_2$H$_2$-type tandem zinc finger proteins (ZFPs) of different lengths, comprising 3 (ZFP$^3$) to 15 (ZFP$^{15}$) zinc finger (ZF) domains. (B) The binding specificity of each zinc-finger domain for ZFPs comprising 3 (ZFP$^3$) to 6 (ZFP$^6$) ZF domains. The colours in each panel correspond to the five selected proteins shown in Fig 2.

binding specificity. The graph shows that, similarly to asymmetrical non-specific binding, asymmetrical specific binding is also common among tethered ZFPs and many of them possess ZF pairs with high and low specificity. Here, the ZF domain with greater DNA binding specificity would contain residues with a higher propensity in most of its specific binding positions (-1, +2, +3 and +6, S1 Fig) whereas the ZFs with lower DNA binding specificity would have other residues in those positions. Defining asymmetry using different $\sigma^{spec}$ values (0.3, 0.4, 0.5 in the second condition) produced similar trends (S2 Fig). We note that many proteins in each length category do not have any asymmetric pairs (0% asymmetry in Fig 4B).

## Cellular abundance of C$_2$H$_2$ type ZFPs varies with their degree of asymmetry

The abundance of a protein can affect its cellular function. To better understand the functional role played by asymmetry in non-specific and specific binding affinities, we examine whether the degree of asymmetry correlates with the protein abundance level. We first analyse the cellular expression level of all 237 ZFPs in the curate database. The distribution of mean abundance for ZFPs of length 3–15 domains indicates that the longer proteins are generally less abundant in cells than proteins with fewer ZF domains, although both high and low abundance proteins can occur for a particular length (Fig 6). For example, the constitutive transcription factor Sp1 (abundance value 23.0) and the inducible transcription factor Egr1 (abundance value 0.2) both contain 3 ZF domains.

To investigate the impact that the cellular expression level of a ZFP may have on its function, we looked for linkage between the degree of asymmetry and abundance. To do so, we divided the proteins into two groups: 1) those exhibiting lower asymmetry (<50% of ZF domain pairs are asymmetric) and 2) those exhibiting higher asymmetry (≥50% of ZF domain pairs are asymmetric). The symmetry groups were examined in the context of both the non-specific and specific DNA binding. The proteins with lower asymmetry are found to be more abundant (Fig 7). This is found for ZFPs having different numbers of ZF domains. Furthermore, this observation is valid for both specific and non-specific binding, although we note that the negative correlation between asymmetry and abundance is stronger in non-specific compared with specific binding (black versus grey bars in Fig 7). The correlation coefficient

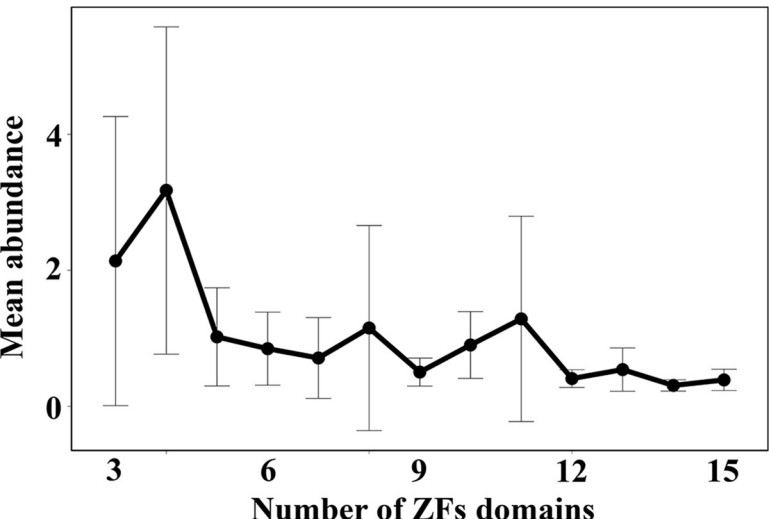

**Fig 6. Abundance of tandem zinc finger proteins of different lengths.** The average abundance is shown as a function of the number of zinc finger domains (3–15) in the proteins.

between asymmetry and abundance is lower for non-specific than specific asymmetry (correlation coefficient of about -0.25 for the nonspecific compared to about -0.08 for the specific asymmetry independently of the number of zinc finger repeat and the cut-off used to define asymmetry; see S2 Table). Furthermore, in the case of non-specific binding of ZFP[3-6] (Fig 7), an unpaired t-test showed that the mean abundance of the symmetric group (2.33) is significantly higher than the asymmetric group mean (0.83) at 95% confidence level (p-value = 0.006). However, the mean abundance is not significantly different for specific binding, 1.92 and 1.85 (p-value = 0.943) for symmetric and asymmetric groups, respectively. A similar trend was observed when asymmetry was calculated with different non-specific and specific cut-offs (see S3 Table).

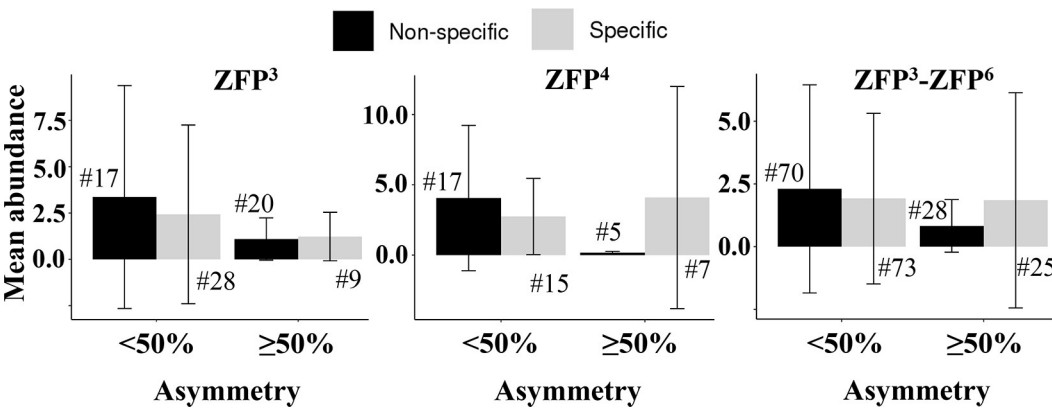

**Fig 7. Relationship between protein abundance and asymmetry.** The zinc-finger proteins (ZFPs) are divided into two groups depending on the percentage of asymmetric zinc finger pairs in each protein: symmetric zinc-finger proteins (containing <50% asymmetric pairs) and asymmetric zinc-finger proteins (containing ≥50% asymmetric pairs). This classification was performed separately for non-specific binding (on the basis of electrostatic net charge) and specific binding (on the basis of the specificity score). The mean abundance of each group is shown by the bar plot. The analysis was performed separately for zinc-finger proteins with 3 (left), 4 (middle), and 3–6 (right) zinc-finger domains.

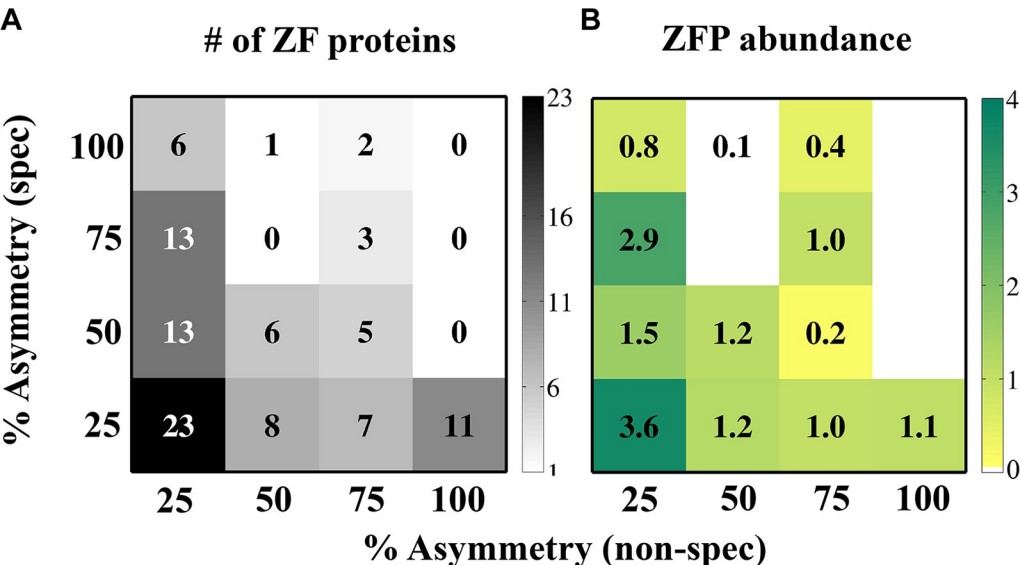

**Fig 8. Percentage of non-specific and specific binding asymmetry in zinc finger proteins (ZFPs) and their linkage with cellular abundance.** Plots of the percent non-specific asymmetry versus the percent specific asymmetry between adjacent zinc finger domains are presented for all ZFPs of length 3–6 domains (ZFP[3-6], 98 proteins). The analysed ZFPs were binned into 16 bins on the basis of the percentage of their non-specific and specific asymmetry scores. The number of ZFPs (grey colour bar) in each of the bins is shown in (A). The mean cellular abundances (yellow-to-green colour bar) of all the ZFPs in each of the bins is shown in (B). This analysis was performed using $\sigma^{nonspec} = 3$ and $\sigma^{spec} = 0.2$.

To better understand the relationship between asymmetry and abundance, we mapped abundance as a function of the non-specific and specific binding asymmetries for proteins with 3–6 ZF domains (Fig 8). Here, the abundance values are represented by a colour scale and mapped onto the percentage of ZF pairs that are asymmetric as defined for non-specific (x-axis) and specific (left y-axis) DNA binding. We binned the ZFP dataset into 16 bins depending on their symmetry scores in each binding context. About 30% of the ZFPs in our dataset possess both non-specific and specific symmetricity whereas 70% of the ZFPs possess some degree of asymmetry (Fig 8A).

Fig 8 shows that the most abundant ZFPs are those with the lowest non-specific and specific asymmetry (see bottom left bin of both matrices). ZFPs with low non-specific asymmetry but varied specific asymmetry are more abundant than ZFPs with higher non-specific asymmetry and low specific asymmetry. Accordingly, similarly to the analysis presented in Fig 7, this approach also shows that the negative correlation between asymmetry and abundance is stronger for non-specific binding than for specific binding (Fig 8). Data for the other ZFP length groups (ZFP[3], ZFP[4] etc.) (S3 Fig) and for different asymmetry cut-off values (S4 Fig) reveal a similar trend.

Finally, Fig 9 displays a few examples of specific proteins from the ZFP[3], ZFP[4], and ZFP[5] length groups to illustrate the relationship between asymmetry in a ZFP and its cellular abundance (Fig 9). In each of these examples, the more highly abundant ZFP (namely, Sp1, transcriptional repressor protein YY1, and E3 ubiquitin-protein ligase ZFP91) has lower electrostatic asymmetry whereas the less abundant ZFP (namely, Egr1, ZFP589, and neurotrophin receptor-interacting factor homolog) has greater electrostatic asymmetry. Taken together, these observations suggest that, compared with those in highly abundant ZFPs, the tandem ZF domains in less abundant human ZFPs may have adapted to become more asymmetric with respect to their electrostatics, and vice versa.

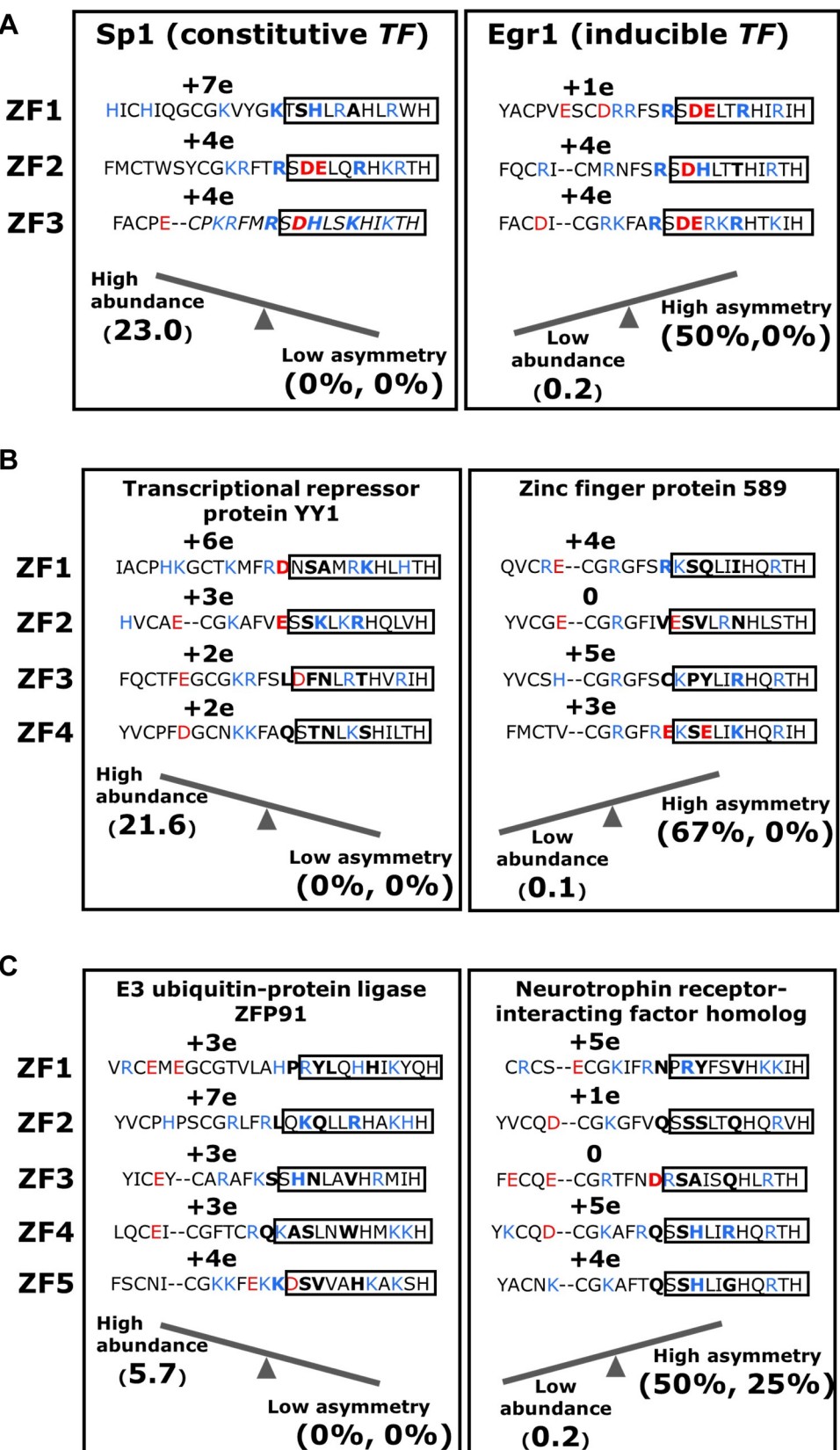

**Fig 9.** Examples of $C_2H_2$-type zinc-finger proteins with low (left side of each panel) and high (right side of each panel) asymmetry. (A) The amino-acid sequences and net charges of the three zinc-finger domains of the constitutive transcription factor Sp1 (left) and the inducible transcription factor Egr-1 (right). The amino-acid sequence is shown for each zinc finger domain (as identified at the left of the box), with the net charge of each domain shown immediately above it. In the sequence, positively and negatively charged residues are coloured in blue and red, respectively, and shown in bold. The recognition helix sequences are enclosed in a box. The positions (-1, +2, +3 and +6) of the four residues involved in specific DNA binding are shown in bold. With respect to the net charge, a pair of adjacent zinc finger domains is considered electrostatically asymmetric if it bears a negative net charge ($<2e$) and/or if the difference in net charge between the two members of the pair is $\geq 3e$. If all pairs of adjacent domains fail these criteria, then the electrostatic asymmetry of the zinc finger protein is 0%. At the bottom of each box, the abundance and the percent asymmetry values are given at the left and right ends, respectively, of the see-saw, which represents their relationship (*i.e.* negative correlation). The two asymmetry percentages shown refer to non-specific binding and specific binding, respectively. Panels (B) and (C) show examples for 4-domain and 5-domain zinc finger proteins, respectively.

## Functional implications of the relationship between asymmetry and abundance in zinc finger proteins

The presence of asymmetry in proteins comprised of tethered ZF domains is widespread and found in proteins of different lengths. Asymmetry is found not only in the non-specific binding context but also in specific binding. The correlation between the degree of non-specific symmetry and the cellular abundance of the protein (Fig 9) may further support the notion that this trend is linked to protein function.

A key property of the interactions of $C_2H_2$-ZFPs with DNA that can be affected by the linkage between non-specific (electrostatic) symmetry and protein abundance is the kinetics of their association (Fig 10). Like many other DNA-binding proteins, recognition of the specific DNA binding site requires that the protein search the genomic DNA (or the relevant accessible DNA). This process, even when restricted to a fraction of the genome, may require scanning thousands of base pairs. High non-specific asymmetry may suggest that only some of the ZF domains engage in a high affinity interaction with non-specific DNA whereas other domains interact more weakly. The ZF domains that are weakly bound to DNA may even dissociate from it. Having some of the ZF domains interacting weakly with DNA may result in a faster search, as the linear diffusion of the ZFP is expected to be faster when the protein–DNA interface during searching is smaller. Non-specific asymmetry between neighbouring ZF domains can facilitate searching also via the monkey-bar mechanism. A ZF domain with lower affinity has a higher probability of interacting with a distant DNA and thus of promoting scanning via monkey-bar jumping. However, non-specific asymmetry requires ZFPs to undergo a conformational change to recognize their target DNA binding site to engage in specific binding.

The kinetics of DNA recognition by symmetric ZFPs is expected to be slower than that of their asymmetric counterparts and the two types tend to adopt different search mechanisms (Fig 10). The effect of asymmetry on the biophysical properties of the sliding and monkey-bar mechanisms was shown experimentally for the Egr-1 transcription factor. Egr-1 is a ZFP with three ZF domains in which the first finger has low affinity to non-specific DNA. In a series of mutants in which the symmetry was increased or even decreased, a clear linkage between search speed and degree of asymmetry was obtained [14]. We note that an excess of ZFs was reported to negatively affect the activity of zinc-finger nucleases, presumably due to slower kinetics which originates from the large interface they form with DNA[22]. This kinetic consequence of the long ZFP is reminiscent to the slow sliding of symmetric ZFPs that can be overcome by higher abundance.

Some wild-type ZFPs possess non-specific symmetricity [8,11,37]. The advantage of symmetric ZFPs is their fast kinetics in recognizing their target site once it is identified. One may wonder how symmetric ZFPs overcome or avoid the slow search kinetics that are a direct

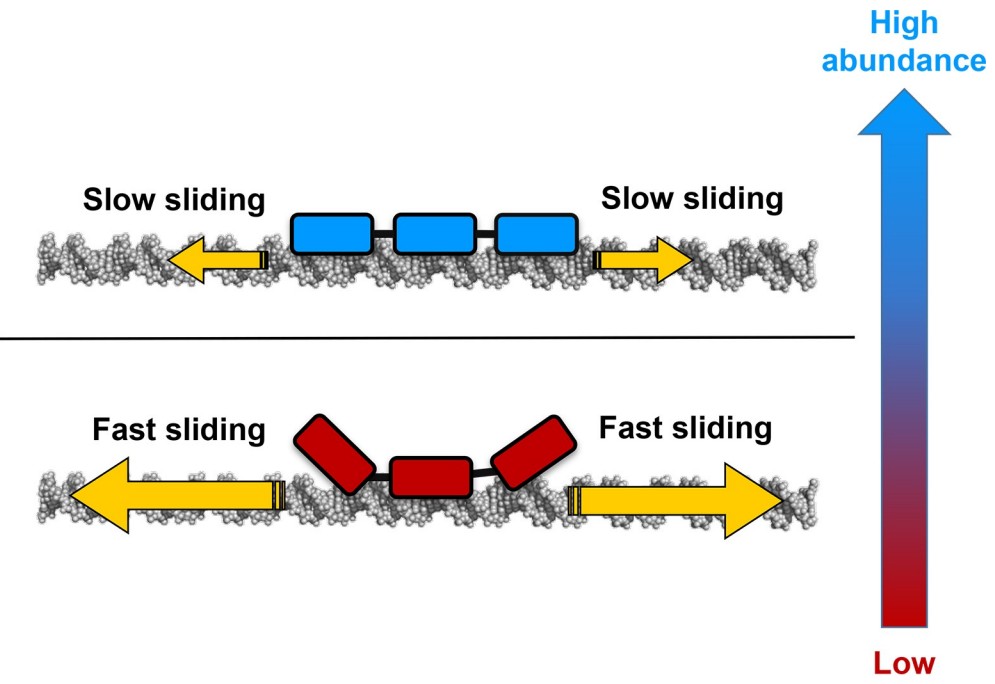

**Fig 10. A schematic illustration on the linkage between asymmetry and abundance and its consequence on recognition kinetics.** The finding that zinc finger proteins with greater non-specific asymmetry are less abundant than protein with lower non-specific asymmetry can be linked to their kinetic of DNA recognition. Proteins with lower asymmetry are expected to diffuse more slowly on DNA, which can be compensated by their higher abundance that increases the probability of fruitful target site recognition. Proteins with greater asymmetry diffuse faster and therefore may search the DNA efficiently even when their abundance is low.

consequence of their symmetry. This drawback can be overcome simply by increasing ZFP concentration in the cell. A larger number of copies of the searching protein will increase the search rate. Indeed, the current study shows that ZFPs with greater symmetry are more abundant in the cell (Fig 8).

## Discussion

In this study, we investigated the linkage between the degree of asymmetry in the tethered domains of multi-domain DNA-binding proteins and their cellular abundance. Asymmetry can be quantified by comparing various biophysical properties of each of the constituent domains. The properties of interest in this study were electrostatic charge and binding specificity, with the degree of asymmetry in their distributions across ZF domains expressed as their non-specific and specific binding asymmetries, respectively. We applied our study to a dataset of 273 human $C_2H_2$-ZFP comprised of 3–15 tandem ZF domains. Focusing on ZF domains is advantageous because they are a common motif that permits statistical analysis as well as having a relatively simple interface with DNA.

Human $C_2H_2$-ZFPs exhibit different degrees of non-specific asymmetry. Some ZFPs show a high degree of electrostatic symmetry between the neighbouring domains, indicating that all the domains interact with non-specific DNA with high affinity. However, many other ZFPs include some ZF domains with lower non-specific affinity to DNA, while their neighbouring domains have higher non-specific affinity. We found that the ZFPs with higher non-specific symmetry are also more abundant in the cell. ZFPs characterized by lower non-specific

symmetry are less abundant in the cell, irrespective of their degree of specific symmetry. Ensuring greater cellular expression levels of ZFPs with high non-specific symmetry could be a means by which to mitigate their expected low diffusion rate along DNA while searching for the target site. This linkage between symmetry and cellular expression level is found for ZFPs of various lengths. For Egr-1 and Sp-1, which are inducible and constitutive transcription factors, respectively, with three ZF domains, the abundance of the latter is three times greater than that of the former. The greater abundance of Sp-1 correlates with its greater non-specific symmetry compared with that of Egr-1 (Fig 9) [11]. The high asymmetry in non-specific binding affinities of Egr-1 were confirmed by both NMR and coarse-grained simulations [37].

We speculate that this relationship between the degree of asymmetry in ZFPs and their cellular abundance will be valid for other multi-domain DNA-binding proteins, as it provides a mechanism for achieving fast scanning of the DNA even when the interface between the protein and DNA is extensive and tight (Fig 10). The reported relationship highlights the role asymmetry may play in protein–DNA interactions and may serve as a design principle of transcription factors.

## Supporting information

**S1 Table. List of 237 tandem $C_2H_2$ type ZFPs used in this study.**
(DOCX)

**S2 Table. Asymmetry-Abundance anti-correlation.** Correlation coefficient between asymmetry and abundance. Data trend shows that asymmetry and abundance are better anti-correlated in non-specific interactions compared to specific interactions, though the correlation-coefficients are low.
(DOCX)

**S3 Table. Difference in abundance of symmetric and asymmetric ZFPs ($<50\%$ and $\geq50\%$ asymmetry, respectively).** A statistical significance by t-test. Data in each cell: $<$symmetric abundance$>$, $<$asymmetric abundance$>$ (p-value). Symmetric group ZFPs has higher mean abundance than the asymmetric group in all cut-off. The difference between symmetric and asymmetric non-specific ZFP binding to DNA are significantly different.
(DOCX)

**S1 Fig. Percentage of amino acid residues found at each position in the $C_2H_2$-type zinc-finger protein (ZFP) sequence.** The propensity of each amino acid residue to locate at each position in the ZF sequence was calculated from multiple sequence alignment of 1911 $C_2H_2$-type ZFP sequences using the sequence of the first finger of the human early growth response protein 1 (Egr1, pdb id 4X9J) as a template. The last column corresponds to the probability of finding a gap at each position.
(TIF)

**S2 Fig. Percentage of non-specific and specific binding asymmetry in zinc finger proteins (ZFPs).** Plots similar to those in Fig 4 are shown for different $\sigma^{nonspec}$ and $\sigma^{spec}$.
(TIF)

**S3 Fig. Plots of the percent specific asymmetry versus the percent non-specific asymmetry.** The presented analysis is for adjacent zinc finger domains in ZFPs comprising 3, 4, 5, 3–4, 3–5 and 4–6 zinc-finger domains (as indicated in the title of each plot). The analysed ZFPs were binned into 16 bins on the basis of the percentage of their non-specific and specific asymmetry scores. The number of ZFPs in each of the bins is shown with grey colour bar. The mean cellular abundances of all the ZFPs in each of the beans bins is shown with yellow-to-green colour

bar. This analysis was performed using $\sigma^{\text{nonspec}} = 3$ and $\sigma^{\text{spec}} = 0.2$.
(TIF)

**S4 Fig. Effect of asymmetry cut-off values on binding asymmetry (specific and non-specific) in zinc finger proteins.** The net charge cut-off value used to define asymmetrical non-specific binding ($\sigma^{\text{nonspec}}$) and the binding specificity score used to define asymmetrical specific binding ($\sigma^{\text{spec}}$) are shown at the top of each panel. Plots similar to those in Fig 8 are shown for the dataset of 98 zinc finger proteins containing 3–6 zinc finger domains. The number of ZFPs in each of the bins is shown with grey colour bar. The mean cellular abundances of all the ZFPs in each of the beans bins is shown with yellow-to-green colour bar.
(TIF)

# Acknowledgments

YL is The Morton and Gladys Pickman professional chair in Structural Biology.

# Author Contributions

**Conceptualization:** Arumay Pal, Yaakov Levy.

**Data curation:** Arumay Pal.

**Formal analysis:** Arumay Pal, Yaakov Levy.

**Project administration:** Yaakov Levy.

**Supervision:** Yaakov Levy.

**Visualization:** Yaakov Levy.

**Writing – original draft:** Arumay Pal.

**Writing – review & editing:** Yaakov Levy.

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
