## [Decision Letter · Decision Letter 0]

28 Jan 2020

Dear Dr. Levy,

Thank you very much for submitting your manuscript "Balance between asymmetry and abundance in multi-domain DNA-binding proteins may regulate the kinetics of their binding to DNA" for consideration at PLOS Computational Biology. As with all papers reviewed by the journal, your manuscript was reviewed by members of the editorial board and by several independent reviewers. The reviewers appreciated the attention to an important topic. Based on the reviews, we are likely to accept this manuscript for publication, providing that you modify the manuscript according to the review recommendations.

Sincerely,

Shi-Jie Chen

Associate Editor

PLOS Computational Biology

Nir Ben-Tal

Deputy Editor

PLOS Computational Biology

[LINK]

Reviewer's Responses to Questions

**Comments to the Authors:**

Reviewer #1: In this manuscript, Pal and Levy describe a statistical analysis of Cys2-His2-type zinc fingers (ZFs) of human transcription factors. In their previous studies, the Levy group conducted coarse-grained simulations of DNA search by ZF proteins and many other classes of transcription factors and found that asymmetry in binding affinity among DNA-binding domains in the same protein might be important for the search kinetics. Based upon the previous findings, the authors examined whether the importance of the asymmetry in the search kinetics is supported by statistical data of human ZF proteins. The authors assessed the asymmetry based on the overall charge distribution among ZFs and the amino-acid types of the key residues for the sequence-specific binding to DNA. Interestingly, ZF proteins with low asymmetry tend to be abundant. Since rapid search kinetics would be nonessential for abundant proteins, the statistical anti-correlation between the asymmetry and abundance seems to support the importance of the asymmetry in the search kinetics. I would like to recommend publication of this manuscript after minor revision in the following issues.

1. In my opinion, the most important result in this work is the data shown in Figure 8. However, the current presentation of this result seems too qualitative. More quantitative information about the statistical significance of the anti-correlation between the asymmetry and the abundance is desirable.

2. The PaxDb database provides abundance data for various cell types. Depending on cell types, the expression levels of individual ZF proteins could be quite different. Which cell types did the authors use for the PaxDb data? Do the authors get the same conclusion for different cell types?

3. It would be nice if the authors could present a graphical scheme to explain why the asymmetry and the abundance could be related in terms of search kinetics. Although there is a nice description in the text, a figure on this concept would be helpful for readers who are not familiar with the previous work of the Levy group.

Minor issues

In Page 3, ‘FoKI’ should be corrected to ‘FokI’.

Reviewer #2: This is a very interesting and well-written article. It is essentially ready for publication, with the following minor optional comments which the authors can be trusted to implement if they wish to do so:

1) This work investigated the linkage between the degree of asymmetry in the tethered domains of multi-domain DNA-binding proteins and their cellular abundance. The dynamics of binding site search per se was not investigated. It is very plausible that the effect found here will indeed be related to the kinetics of binding and the dynamics of binding site search, but as of now this is just a hypothesis. I think it is a bit too much to mention kinetics in the title of the manuscript. I’d rather focus the title on the solid results obtained here.

2) The terms such as non-specific asymmetry used in the abstract needs to be better explained because it is the central point of the article.

3) Methods section: how many proteins were included in the final dataset with defined abundance and asymmetry?

4) Methods section: I would add a dedicated paragraph explaining the asymmetry and all calculations of the specific and non-specific asymmetry. This explanation is provided later in the results section but it would be good to have it also in the Methods in a condensed form with equation.

5) How was the threshold value selected for specific and non-specific asymmetry?

6) Figure 4: May be make violin plots rather than bar-plots, to show the distribution of values?

7) Figure 7: What is indicated by the dark rectangle?

8) The protein abundance used throughput the article is cell-type specific. It is important to indicate the cell type. This may be mentioned in several places including figure legends, e.g. Figure 7.

**Have all data underlying the figures and results presented in the manuscript been provided?**

Reviewer #1: None

Reviewer #2: Yes

PLOS authors have the option to publish the peer review history of their article (what does this mean?). If published, this will include your full peer review and any attached files.

Reviewer #1: No

Reviewer #2: No
---

## [Decision Letter · Decision Letter 1]

11 Apr 2020

Dear Dr. Levy,

We are pleased to inform you that your manuscript 'Balance between asymmetry and abundance in multi-domain DNA-binding proteins may regulate the kinetics of their binding to DNA' has been provisionally accepted for publication in PLOS Computational Biology.

Best regards,

Shi-Jie Chen

Associate Editor

PLOS Computational Biology

Nir Ben-Tal

Deputy Editor

PLOS Computational Biology

Reviewer's Responses to Questions

**Comments to the Authors:**

Reviewer #1: The authors have revised the manuscript appropriately. The issues raised for the initial submission have been addressed. I don't have any other concerns. So, I recommend publication of this manuscript as is.

**Have all data underlying the figures and results presented in the manuscript been provided?**

Reviewer #1: Yes

PLOS authors have the option to publish the peer review history of their article (what does this mean?). If published, this will include your full peer review and any attached files.

Reviewer #1: No

---

## [Editor Report · Acceptance letter]

13 May 2020

PCOMPBIOL-D-19-02058R1 

Balance between asymmetry and abundance in multi-domain DNA-binding proteins may regulate the kinetics of their binding to DNA

Dear Dr Levy,

I am pleased to inform you that your manuscript has been formally accepted for publication in PLOS Computational Biology. Your manuscript is now with our production department and you will be notified of the publication date in due course.

With kind regards,

Laura Mallard
